# Near-Optimal Policies for Dynamic Multinomial Logit Assortment Selection Models

**Yining Wang**
Machine Learning Department
Carnegie Mellon University
yiningwa@cs.cmu.edu

**Xi Chen**
Stern School of Business
New York University
xchen3@stern.nyu.edu

**Yuan Zhou**
Computer Science Department
Indiana University at Bloomington
and
Department of Industrial and Enterprise Systems Engineering
University of Illinois at Urbana-Champaign
yuanz@illinois.edu

## Abstract

In this paper we consider the dynamic assortment selection problem under an uncapacitated multinomial-logit (MNL) model. By carefully analyzing a revenue potential function, we show that a trisection based algorithm achieves an *item-independent* regret bound of $O(\sqrt{T \log\log T})$, which matches information theoretical lower bounds up to iterated logarithmic terms. Our proof technique draws tools from the unimodal/convex bandit literature as well as adaptive confidence parameters in minimax multi-armed bandit problems.

**Keywords**: dynamic assortment planning, multinomial logit choice model, trisection algorithm, regret analysis.

## 1 Introduction

Assortment planning has a wide range of applications in e-commerce and online advertising. Given a large number of substitutable products, the assortment planning problem refers to the selection of a subset of products (a.k.a., an assortment) offering to a customer such that the expected revenue is maximized [2, 3, 14, 17, 20]. Given $N$ items, each associated with a revenue parameter [1] $r_i \in [0, 1]$ representing the revenue a retailer collects once a customer purchases the $i$-th item. The revenue parameters $\{r_i\}_{i=1}^N$ are typically known to the retailer, who has full knowledge of each item's prices/costs. In a dynamic assortment planning problem, assuming that there are a total of $T$ time epochs, the retailer presents an *assortment* $S_t \subseteq [N]$ to an incoming customer, and observes his/her purchasing action $i_t \in S_t \cup \{0\}$. (If $i_t = 0$ then the customer makes no purchases at time $t$.) If a purchasing action is made (i.e., $i_t \neq 0$), the corresponding revenue $r_{i_t}$ is collected. It is worthy noting that since items are substitutable (e.g., different models of cell phones), a typical setting of assortment planning usually restricts a purchase to be a single item.

The retailer's objective is to maximize the expected revenue over the $T$ time periods. Such objectives can be best measured and evaluated under a "regret minimization" framework, in which the retailer's

assortment sequence is compared against the optimal assortment. More specifically, consider

$$\text{Regret}(\{S_t\}_{t=1}^T) := \mathbb{E} \sum_{t=1}^T R(S^*) - R(S_t), \quad S^* \in \arg\min_{S \subseteq [N]} R(S) \qquad (1)$$

as the regret measure of an assortment sequence $\{S_t\}_{t=1}^T$, where $R(S_t) = \mathbb{E}[r_{i_t}|S_t]$ is the expected revenue the retailer collects on assortment $S_t$ (for notational convenience we define $r_0 = 0$ corresponding to the "no-purchase" action).

For the regret measure Eq. (1) to be well-defined, it is conventional to specify a probabilistic model (known as "choice model") that governs a customer's purchasing choice $i_t \in S_t \cup \{0\}$ on a provided assortment $S_t$. Perhaps the most popular choice model is the *multinomial-logit (MNL)* choice model [5, 18, 22], which assigns each item $i \in [N]$ a "preference parameter" $v_i \geqslant 0$ and the purchasing choice $i_t \in S_t \cup \{0\}$ is modeled by

$$\Pr[i_t = j|S_t] = \frac{v_j}{v_0 + \sum_{k \in S_t} v_k}, \qquad \forall j \in S_t \cup \{0\}. \qquad (2)$$

Subsequently, the expected revenue $R(S_t)$ can be expressed as

$$R(S_t) = \sum_{j \in S_t} r_j \Pr[i_t = j|S_t] = \frac{\sum_{j \in S_t} r_j v_j}{v_0 + \sum_{j \in S_t} v_j}. \qquad (3)$$

For normalization purposes the preference parameter for the "no-purchase" action is assumed to be $v_0 = 1$. Apart from that, the rest of the preference parameters $\{v_i\}_{i=1}^N$ are *unknown* to the retailer and have to be either explicitly or implicitly learnt from customers' purchasing actions $\{i_t\}_{t=1}^T$.

## 1.1 Our results and techniques

The main contribution of this paper is an optimal characterization of the worst-case regret under the MNL assortment selection model specified in Eqs. (1) and (2). More specifically, we have the following informal statement of the main results in this paper.

**Theorem 1** (informal). *There exists a policy whose worst-case regret over $T$ time periods is upper bounded by $C_1 \sqrt{T \log \log T}$ for some universal constant $C_1 > 0$; furthermore, there exists another universal constant $C_2 > 0$ such that no policy can achieve worst-case regret smaller than $C_2 \sqrt{T}$.*

An important aspect of Theorem 1 is that our regret bound is completely *independent of the number of items $N$*, which improves the existing dynamic regret minimization results on the MNL assortment selection problem [2, 3, 20]. This property makes our result more favorable for scenarios when a large number of potential items are available, e.g., online sales or online advertisement.

To enable such an $N$-independent regret, we provide a refined analysis of a certain *unimodal* revenue potential function first studied in [20] and consider a trisection algorithm on revenue levels, borrowing ideas from literature on unimodal bandits on either discrete or continuous arm domains [1, 11, 23]. An important challenge is that the revenue potential function (defined in Eq. (4)) does not satisfy convexity or local Lipschitz growth, [2] and therefore previous results on unimodal bandits cannot be directly applied. On the other hand, it is a simple exercise that mere unimodality in multi-armed bandits cannot lead to regret smaller than $\sqrt{NT}$, because the worst-case constructions in the classical lower bound or multi-armed bandits have unimodal arms [6, 7]. To overcome such difficulties, we establish additional properties of the potential function in Eq. (4) which are different from classical convexity or Lipschitz growth properties. In particular, we prove connections between the potential function and the straight line $F(\theta) = \theta$, which is then used as guidelines in our update rules of trisection. Also, because the potential function behaves differently on $F(\theta) \leqslant \theta$ and $F(\theta) \geqslant \theta$, our trisection algorithm is *asymmetric* in the treatments of the two trisection mid-points, which is in contrast to previous trisection based methods for unimodal bandits [11, 23] that treat both trisection mid-points symmetrically.

We also remark that the upper and lower bounds in Theorem 1 match except for an $\log \log T$ term. Under the "gap-free" setting where $O(\sqrt{T})$ regret is to be expected, the removal of additional $\log T$

terms in dynamic assortment selection and unimodal bandit problems is highly non-trivial. Most previous results on dynamic assortment selection [2, 3, 20] and unimodal/convex bandit [1, 11, 23] have additional $\log T$ terms in regret upper bounds. (The work of [11] also derived gap-dependent regret bounds for unimodal bandit, which is not easily comparable to our bounds.) The improvement from $\log T$ to $\log \log T$ achieved in this paper is done by using a sharper law-of-the-iterated-logarithm (LIL) type concentration inequalities [16] and an adaptive confidence strategy similar to the MOSS algorithm for multi-armed bandits [4]. Its analysis, however, is quite different from the analysis of the MOSS algorithm in [4] and also yields an additional $\log \log T$ factor. We conjecture that the additional $\log \log T$ factor can also be removed by resorting to much more complicated procedures, as we discuss in Sec. 6.

## 1.2 Related work

The question of *dynamic* optimization of commodity assortments has received increasing attention in both the machine learning and operations management society [2, 3, 8, 19, 21], as the mean utilities of customers (corresponding to the preference parameters $\{v_i\}$ in our model) are typically unknown and have to be learnt on the fly.

The work of [19] is perhaps the closest to our paper, which analyzed the same revenue potential function and designed a golden-ratio search algorithm whose regret only depends logarithmically on the number of items. The analysis of [19] assumes a constant gap between *any two* assortment level sets, which might fail to hold when the number of items $N$ is large. In this work we relax the gap assumption and also remove the additional $\log N$ dependency by a more refined analysis of properties of the revenue potential function and borrowing "trisection" ideas from the unimodal bandit literature [1, 11, 23].

The works of [2, 3] considered variants of UCB/Thompson sampling type methods and focused primarily on the *capacitated* MNL assortment model, in which the size of each assortment $S_t$ is not allowed to exceed a pre-specified parameter $K < N$. It is known that the regret behavior in capacitated and uncapacitated models can be vastly different: in the capacitated case a $\sqrt{NT}$ regret lower bound exists provided that $K < N/4$, while for the uncapacitated model it is possible to achieve $\log N$ or even $N$-independent regret.

Another relevant line of research is *unimodal bandit* [1, 11, 12, 23], in which discrete or continuous multi-armed bandit problems are considered with additional unimodality constraints on the means of the arms. Apart from unimodality, additional structures such as "inverse Lipschitz continuity" (e.g., $|\mu(i) - \mu(j)| \geqslant L|i - j|$) or convexity are imposed to ensure improvement of regret, both of which fail to hold for the potential function $F$ arising from uncapacitated MNL assortment choice problems. In addition, under the "gap-free" setting where an $O(\sqrt{T})$ regret is to be expected, most previous works have additional $\log T$ terms in their regret upper bounds, except for the work of [12] which introduces additional strong regularity conditions on the underlying functions.

In [10], a more general problem of optimizing piecewise-constant function is considered, without unimodal structure of the function assumed. Consequently, a weaker $\widetilde{O}(T^{2/3})$ regret is derived.

## 2 The revenue potential function and its properties

For the MNL assortment selection model without capacity constraints, it is a classical result that the optimal assortment must consist of items with the largest revenue parameters (see, e.g., [17]):

**Proposition 1.** *There exists $\theta \in [0, 1]$ such that $\mathcal{L}_\theta := \{i \in [N] : r_i \geqslant \theta\}$ satisfies $R(\mathcal{L}_\theta) = R(S^*)$.*

Proposition 1 suggests that it suffices to consider "level-set" type assortments $\mathcal{L}_\theta = \{i \in [N] : r_i \geqslant \theta\}$ and finds $\theta \in [0, 1]$ that gives rises to the largest $R(\mathcal{L}_\theta)$. This motivates the following "potential" function, which takes a revenue threshold $\theta$ as input and outputs the expected revenue of its corresponding level set assortments:

$$\text{The revenue potential function:} \quad F(\theta) := R(\mathcal{L}_\theta), \ \ \theta \in [0, 1]. \tag{4}$$

The potential $F$ was first introduced and considered in [17], in which it was proved that $F$ is left-continuous, piecewise-constant and *unimodal* in its input revenue $\theta$. Using such unimodality, a

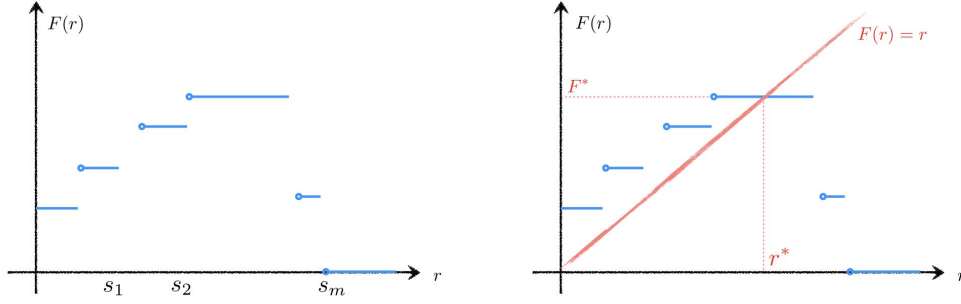

Figure 1: Illustration of the potential function $F(\theta)$, the important quantities $F^*$ and $\theta^*$, and their properties.

golden-ratio search based policy was designed that achieves $O(\log N \log T)$ regret under additional consecutive gap assumptions of the level set assortments $\{\mathcal{L}_\theta\}$. To derive gap-independent results and to get rid of the additional $\log N$ dependency, we provide a more refined analysis of properties of the potential function $F$ in this paper, summarized in the following three lemmas:

**Lemma 1.** *There exists $\theta^* > 0$ such that $\theta^* = F(\theta^*) = F^* = \sup_{\theta \geqslant 0} F(\theta) = R(S^*)$.*

**Lemma 2.** *For any $\theta \geqslant \theta^*$, $F(\theta) \leqslant \theta$ and $F(\theta) \geqslant F(\theta^+)$, where $F(\theta^+) = \lim_{\varphi \to \theta^+} F(\varphi)$.*

**Lemma 3.** *For any $\theta \leqslant \theta^*$, $F(\theta) \geqslant \theta$ and $F(\theta) \leqslant F(\theta^+)$.*

The proofs of the above lemmas are given in the appendix. The give a rather complete picture of the behavior of the potential function $F$, and most importantly the relationship between $F$ and the central straight line $F(r) = r$, as depicted in Figure 1. More precisely, The mode of $F$ occurs at its intersection with $F(r) = r$ and monotonically decreases moving away from $\theta^*$ in both directions. This helps us gauge the positioning of a particular revenue level $\theta$ by simply comparing the expected revenue of $R(\mathcal{L}_\theta)$ with $\theta$ itself, motivating an asymmetric trisection algorithm which we describe in the next section.

## 3  Trisection and regret analysis

We propose an algorithm based on trisections of the potential function $F$ in order to locate level $\theta^*$ at which the maximum expected revenue $F^* = F(\theta^*)$ is attained. Our algorithm avoids explicitly estimating individual items' mean utilities $\{v_i\}_{i=1}^N$, and subsequently yields a regret independent of the number of items $N$. We first give a simplified algorithm (pseudo-code description in Algorithm 1) with an additional $O(\sqrt{\log T})$ term in the regret upper bound and outline its proofs. We further show how the additional dependency on $T$ can be improved to $O(\sqrt{\log \log T})$ and eventually fully removed by using more advanced techniques. Due to space constraints, complete proofs of all results are deferred to the appendix.

To assist with readability, below we list notations used in the algorithm description together with their meanings:

- $a_\tau$ and $b_\tau$: left and right boundaries that contain $\theta^*$; it is guaranteed that $a_\tau \leqslant \theta^* \leqslant b_\tau$ with high probability, and the regret incurred on failure events is strictly controlled;
- $x_\tau$ and $y_\tau$: trisection points; $x_\tau$ is closer to $a_\tau$ and $y_\tau$ is closer to $b_\tau$;
- $\ell_t(y_\tau)$ and $u_t(y_\tau)$: lower and upper confidence bands for $F(y_\tau)$ established at iteration $t$; it is guaranteed that $\ell_t(y_\tau) \leqslant F(y_\tau) \leqslant u_t(y_\tau)$ with high probability, and the regret incurred on failure events is strictly controlled;
- $\rho_t(y_\tau)$: accumulated reward by exploring level set $\mathcal{L}_{y_\tau}$ up to iteration $t$.

With these notations in place, we provide a detailed description of Algorithm 1 to facilitate the understanding. The algorithm operates in epochs (outer iterations) $\tau = 1, 2, \cdots$ until a total of $T$

---

**Input:** revenue parameters $r_1, \cdots, r_n \in [0, 1]$, time horizon $T$
**Output:** sequence of assortment selections $S_1, S_2, \cdots, S_T \subseteq [N]$

1   Initialization: $a_0 = 0$, $b_0 = 1$;
2   **for** $\tau = 0, 1, \cdots$ **do**
3      $x_\tau = \frac{2}{3}a_\tau + \frac{1}{3}b_\tau$, $y_\tau = \frac{1}{3}a_\tau + \frac{2}{3}b_\tau$ ;                       ▷ *trisection*
4      $\ell_0(x_\tau) = \ell_0(y_\tau) = 0$, $u_0(x_\tau) = u_0(y_\tau) = 1$ ;       ▷ *initialization of confidence intervals*
5      $\rho_0(x_\tau) = \rho_0(y_\tau) = 0$ ;                 ▷ *initialization of accumulated rewards*
6      **for** $t = 1$ *to* $16\lceil (y_\tau - x_\tau)^{-2} \ln(T)) \rceil$ [4] **do**
7          **if** $\ell_{t-1}(y_\tau) \leqslant y_\tau \leqslant u_{t-1}(y_\tau)$ **then** $\rho_t(y_\tau), \ell_t(y_\tau), u_t(y_\tau) \leftarrow \text{EXPLORE}(y_\tau, t, 1/T^2)$ ;
8          **else** $\rho_t(y_\tau), \ell_t(y_\tau), u_t(y_\tau) \leftarrow \rho_{t-1}(y_\tau), \ell_{t-1}(y_\tau), u_{t-1}(y_\tau)$;
9          Exploit the left endpoint $a_\tau$: pick assortment $S = \mathcal{L}_{a_\tau}$;
10     **end**
       ▷ *Update trisection parameters*
11     **if** $u_t(y_\tau) < y_\tau$ **then** $a_{\tau+1} = a_\tau$, $b_{\tau+1} = y_\tau$ ;
12     **else** $a_{\tau+1} = x_\tau$, $b_{\tau+1} = b_\tau$;
13   **end**

---

**Algorithm 1:** The trisection algorithm.

assortment selections are made. The objective of each outer iteration $\tau$ is to find the relative position between trisection points $(x_\tau, y_\tau)$ and the "reference" location $\theta^*$, after which the algorithm either moves $a_\tau$ to $x_\tau$ or $b_\tau$ to $y_\tau$, effectively shrinking the length of the interval $[a_\tau, b_\tau]$ that contains $\theta^*$ to its two thirds. Furthermore, to avoid a large cumulative regret, level set corresponding to the left endpoint $a_\tau$ is exploited in each time period within the epoch $\tau$ to offset potentially large regret incurred by exploring $y_\tau$.

In Steps 7 and 8 of Algorithm 1, lower and upper confidence bands $[\ell_t(y_\tau), u_t(y_\tau)]$ for $F(y_\tau)$ are constructed using concentration inequalities (e.g. Hoeffding's inequality [15]). These confidence bands are updated until the relationship between $y_\tau$ and $F(y_\tau)$ is clear, or a pre-specified number of inner iterations for outer iteration $\tau$ has been reached (set to $n_\tau := \lceil 16(y_\tau - x_\tau)^{-2} \ln(T^2) \rceil$ in Step 6). Algorithm 2 gives detailed descriptions on how such confidence intervals are built, based on repeated exploration of level set $\mathcal{L}_{y_\tau}$.

After sufficiently many explorations of $\mathcal{L}_{y_\tau}$, a decision is made on whether to advance the left bounary (i.e., $a_{\tau+1} \leftarrow x_\tau$) or the right boundary (i.e., $b_{\tau+1} \leftarrow y_\tau$). Below we give high-level intuitions on how such decisions are made, with rigorous justifications presented later as part of the proof of the main regret theorem for Algorithm 1.

1. If there is sufficient evidence that $F(y_\tau) < y_\tau$ (e.g., $u_t(y_\tau) < y_\tau$), then $y_\tau$ must be to the right of $\theta^*$ (i.e., $y_\tau \geqslant \theta^*$) due to Lemma 2. Therefore, we will shrink the value of right boundary by setting $b_{\tau+1} \leftarrow y_\tau$.

2. On the other hand, when $u_t(y_\tau) \geqslant y_\tau$, we can conclude that $x_\tau$ *must be to the left of* $\theta^*$ *(i.e.,* $x_\tau \leqslant \theta^*$*)*. We show this by contradiction. Assuming that $x_\tau > \theta^*$, since $y_\tau$ is always greater than $x_\tau$ (and thus $y_\tau > \theta^*$) and the gap between $y_\tau$ and $F(y_\tau)$ is at least $y_\tau - x_\tau$ [3], the gap will be detected by the confidence bands and thus we will have $u_t(y_\tau) < y_\tau$ with high probability. This leads to a contradiction.

 Therefore, since $x_\tau$ is to the left of $\theta^*$, we should increase the value of the left boundary by setting $a_{\tau+1} \leftarrow x_\tau$.

The following theorem is our main upper bound result for the (worst-case) regret incurred by Algorithm 1.

> **Input:** revenue level $\theta$, time $t$, confidence level $\delta$
> **Output:** accumulated revenue $\rho_t(\theta)$, confidence intervals $\ell_t(\theta)$ and $u_t(\theta)$
> 1 Pick assortment $S = \mathcal{L}_\theta(\mathcal{N})$ and observe purchasing action $j \in S \cup \{0\}$;
> 2 Update accumulated reward: $\rho_t(\theta) = \rho_{t-1}(\theta) + r_j$ ; $\qquad\qquad\qquad \triangleright r_0 := 0$
> 3 Update confidence intervals: $[\ell_t(\theta), u_t(\theta)] = \frac{\rho_t(\theta)}{t} \pm \sqrt{\frac{\log(1/\delta)}{2t}}$.

**Algorithm 2:** EXPLORE Subroutine: exploring a certain revenue level $\theta$

**Theorem 2.** *There exists a universal constant $C_1 > 0$ such that for all parameters $\{v_i\}_{i=1}^N$ and $\{r_i\}_{i=1}^N$ satisfying $r_i \in [0, 1]$, the regret incurred by Algorithm 1 satisfies*

$$\mathrm{Reg}(\{S_t\}_{t=1}^T) = \mathbb{E}\sum_{t=1}^T R(S^*) - R(S_t) \leqslant C_1\sqrt{T\log T}. \tag{5}$$

### 3.1 Improved regret with LIL confidence intervals

In this section we consider a variant of Algorithm 1 that achieves an improved regret of $O(\sqrt{T\log\log T})$. The key idea is to use the finite-sample law-of-iterated-logarithm (LIL, [13]) confidence intervals [16] together with an adaptive choice of confidence parameters similar to the MOSS strategy [4] in order to carefully upper bounding regret induced by failure probabilities.

More specifically, most steps in Algorithms 1 and 2 remain unchanged, and the changes we make are summarized below:

- Step 3 in Algorithm 2 is replaced with an LIL-confidence interval [16]:

$$[\ell_t(\theta), u_t(\theta)] = \frac{\rho_t(\theta)}{t} \pm 4\sqrt{\frac{\ln\ln(2T) + \ln(112/\delta)}{t}}. \tag{6}$$

- Step 7 in Algorithm 1 is replaced with EXPLORE$(y_\tau, t, 1/(T(y_\tau - x_\tau)^2))$ for an adaptive confidence parameter $\delta = 1/(T(y_\tau - x_\tau)^2)$; correspondingly, the number of inner iterations is changed to $n_\tau = 64[(y_\tau - x_\tau)^{-2}[\ln\ln(2T) + \ln(112T(y_\tau - x_\tau)^2)]]$

The first change we make to achieve improved regret is the way how confidence intervals $[\ell_t(\theta), u_t(\theta)]$ of $F(\theta)$ is constructed. Comparing the new confidence interval in Eq. (6) with the original one in Algorithm 2, the important difference is the $\ln\ln(2T)$ term arising from the law of the iterated logarithm, which makes the confidence intervals hold *uniformly* for all $t$. This also leads to a different choice of confidence parameter $\delta$ in constructing confidence intervals, which is the second important change we make. In particular, instead of using a universal confidence level [5] $\delta = O(1/T^2)$ throughout the entire procedure, "adaptive" confidence levels $\delta = O(1/(T(y_\tau - x_\tau)^2))$ are used, which increases as the algorithm moves onto later iterations. Such choice of confidence parameters is motivated by the fact that the accumulated regret suffers less from a confidence interval failure at later iterations. Indeed, since we are relatively closer to the optimal assortment, the "excess regret" suffered when the confidence interval fails to cover the true potential function value is smaller. We also remark that similar confidence parameter choices were also adopted in [4] to remove additional $\log(T)$ factors in multi-armed bandit problems.

The following theorem shows that the algorithm variant presented above achieves an asymptotic regret of $O(\sqrt{T\log\log T})$, considerably improving Theorem 2 establishing an $O(\sqrt{T\log T})$ regret bound. Its proof is rather technical and involves careful analysis of failure events at each outer iteration $\tau$ of the trisection algorithm. Due to space constraints, we defer the entire proof of Theorem 3 to the appendix.

**Theorem 3.** *There exists a universal constant $C_1 > 0$ such that for all parameters $\{v_i\}_{i=1}^N$ and $\{r_i\}_{i=1}^N$ satisfying $r_i \in [0,1]$, the regret incurred by the variant of Algorithm 1 satisfies*

$$\text{Reg}(\{S_t\}_{t=1}^T) = \mathbb{E} \sum_{t=1}^T R(S^*) - R(S_t) \leqslant C_1 \sqrt{T \log \log T}. \tag{7}$$

## 4  Lower bound

We prove the following theorem showing that no policy can achieve an accumulated regret smaller than $\Omega(\sqrt{T})$ in the worst case.

**Theorem 4.** *Let $N$ and $T$ be the number of items and the time horizon that can be arbitrary. There exists revenue parameters $r_1, \cdots, r_N \in [0,1]$ such that for any policy $\pi$,*

$$\sup_{v_1, \cdots, v_N \geqslant 0} \mathbb{E} \sum_{t=1}^T R(S^*) - R(S_t) \geqslant \sqrt{T}/384. \tag{8}$$

Theorem 4 shows that our regret upper bounds in Theorems 2 and 3 are tight up to $\sqrt{\log T}$ or $\sqrt{\log \log T}$ factors and numerical constants. We conjecture (in Sec. 6) that the additional $\sqrt{\log \log T}$ term can also be removed, leading to upper and lower bounds that match up to universal constants.

We next give a sketch of the proof of Theorem 4. Due to space constraints, we only present an outline of the proof and defer proofs of all technical lemmas to the appendix.

We first describe the underlying parameter values on which our lower bound proof is built. Fix revenue parameters $\{r_i\}_{i=1}^N$ as $r_1 = 1$, $r_2 = 1/2$ and $r_3 = \cdots = r_N = 0$, which are known a priori. We then consider two constructions of the unknown mean utility parameters $\{v_i\}_{i=1}^N$:

$$P_0: \quad v_1 = 1 - 1/4\sqrt{T}, \ v_2 = 1, \ v_3 = \cdots = v_N = 0;$$
$$P_1: \quad v_1 = 1 + 1/4\sqrt{T}, \ v_2 = 1, \ v_3 = \cdots = v_N = 0.$$

We note that $P_0$ and $P_1$ also give the probability distributions that characterize the customer random purchasing actions; and thus we will use $P_j[A]$ to denote the probability of event $A$ under the utility parameters specified by $P_j$ for $j \in \{0,1\}$.

The first lemma shows that there does not exist estimators that can identify $P_0$ from $P_1$ with high probability with only $T$ observations of random purchasing actions. Its proof involves careful calculation of the Kullback-Leibler (KL) divergence between the two hypothesized distributions and subsequent application of Le Cam's lemma to the testing question between $P_0$ and $P_1$.

**Lemma 4.** *For any estimator $\widehat{\psi} \in \{0,1\}$ whose inputs are $T$ random purchasing actions $i_1, \cdots, i_T$, it holds that $\max_{j \in \{0,1\}} P_j[\widehat{\psi} \neq j] \geqslant 1/3$.*

On the other hand, the following lemma shows that, if the policy $\pi$ can achieve a small regret under both $P_0$ and $P_1$, then one can construct an estimator based on $\pi$ such that with large probability the estimator can distinguish between $P_0$ and $P_1$ from observed customers' purchasing actions.

**Lemma 5.** *Suppose a policy $\pi$ satisfies $\text{Regret}(\{S_t\}_{t=1}^T) < \sqrt{T}/384$ for both $P_0$ and $P_1$. Then there exists an estimator $\widehat{\psi} \in \{0,1\}$ such that $P_j[\widehat{\psi} \neq j] \leqslant 1/4$ for both $j = 0$ and $j = 1$.*

Lemma 5 is proved by explicitly constructing a classifier (tester) $\widehat{\psi}$ from any sequence of low regret. In particular, for any assortment sequence $\{S_t\}_{t=1}^T$, we construct $\widehat{\psi}$ as $\widehat{\psi} = 0$ if $\frac{1}{T} \sum_{t=1}^T \mathbb{I}[1 \in S_t, 2 \notin S_t] \geqslant 1/2$ and $\widehat{\psi} = 1$ otherwise. Using Markov's inequality and the construction of $\{r_i, v_i\}$, it can be shown that if $\text{Regret}(\{S_t\}_{t=1}^T) > \sqrt{T}/384$ then $\widehat{\psi}$ is a good tester with small testing error. Detailed calculations and the complete proof is deferred to the appendix.

Combining Lemmas 4 and 5 we proved our lower bound result in Theorem 4.

## 5  Numerical results

We present simple numerical results of our proposed trisection (and its LIL-improved variant) algorithm and compare their performance with several competitors on synthetic data.

Table 1: Average (mean) and worst-case (max) regret of our trisection and LIL-trisection algorithms and their competitors on synthetic data. $N$ is the number of items and $T$ is the time horizon.

| $(N, T)$ | UCB | | THOMPSON | | GRS | | TRISEC. | | LIL-TRISEC. | |
|---|---|---|---|---|---|---|---|---|---|---|
| | mean | max | mean | max | mean | max | mean | max | mean | max |
| (100,500) | 34.9 | 38.1 | 1.28 | 2.97 | 10.9 | 22.4 | 7.68 | 7.68 | 5.17 | 5.17 |
| (250,500) | 54.3 | 56.2 | 2.81 | 4.95 | 7.93 | 34.2 | 7.57 | 7.57 | 5.02 | 5.02 |
| (500,500) | 73.4 | 75.5 | 4.90 | 4.95 | 7.02 | 43.4 | 7.43 | 7.43 | 4.91 | 4.91 |
| (1000,500) | 90.3 | 93.5 | 8.17 | 10.7 | 5.34 | 45.1 | 7.44 | 7.44 | 4.74 | 4.74 |
| | | | | | | | | | | |
| (100,1000) | 73.1 | 78.2 | 1.36 | 2.79 | 139.9 | 175.0 | 8.69 | 8.69 | 5.36 | 5.36 |
| (250,1000) | 113.7 | 119.3 | 3.36 | 5.17 | 90.1 | 110.1 | 8.69 | 8.69 | 5.31 | 5.31 |
| (500,1000) | 136.8 | 140.3 | 5.65 | 7.64 | 65.7 | 113.9 | 9.38 | 9.38 | 6.01 | 6.01 |
| (1000, 1000) | 160.8 | 165.4 | 9.31 | 12.4 | 8.43 | 22.8 | 9.77 | 9.77 | 6.39 | 6.39 |

**Experimental setup.** We generate each of the revenue parameters $\{r_i\}_{i=1}^N$ independently and identically from the uniform distribution on $[.4, .5]$. For the preference parameters $\{v_i\}_{i=1}^N$, they are generated independently and identically from the uniform distribution on $[10/N, 20/N]$, where $N$ is the total number of items available.

To motivate our parameter setting, consider the following three types of assortments: the "single assortment" $S = \{i\}$ for some $i \in [N]$, the "full assortment" $S = \{1, 2, \cdots, N\}$, and the "appropriate" assortment $S = \{i \in [N] : r_i \geqslant 0.42\}$. For the single assortment $S = \{i\}$, because the preference parameter for each item is rather small ($v_i \leqslant 20/N$), no single assortment can produce an expected revenue exceeding $0.5 \times (20/N)/(1 + 20/N) = 10/(20 + N)$. For the full assortment $S = \{1, 2, \cdots, N\}$, because $\sum_{i=1}^N r_i v_i \xrightarrow{p} 0.45 \times 15/N \times N = 6.75$ and $\sum_{i=1}^N v_i \xrightarrow{p} 15$ by the law of large numbers, the expected revenue of $S$ is around $6.75/(1 + 15) = 0.422$. Finally, for the "appropriate" assortment $S = \{i \in [N] : r_i \geqslant 0.42\}$, we have $\sum_{i \in S} r_i v_i \xrightarrow{p} 0.46 \times 15/N \times 0.8N = 5.52$ and $\sum_{i \in S} v_i \xrightarrow{p} 15/N \times 0.8N = 12$. Therefore, the expected revenue of $S$ is around $5.52/(1 + 12) = 0.425 > 0.422$. The above discussion shows that a revenue threshold $r^* \in (0.4, 0.5)$ is mandatory to extract a portion of the items $\{i \in [N] : r_i \geqslant r^*\}$ that attain the optimal expected revenue, which is highly non-trivial for a dynamic assortment selection algorithm to identify.

**Comparative methods.** Our trisection algorithm with $O(\sqrt{T \log T})$ regret is denoted as TRISEC, and its LIL-variant (with regret $O(\sqrt{T \log \log T})$) is denoted as LIL-TRISEC. The other methods we compare against include the *Upper Confidence Bound* algorithm of [2] (denoted as UCB), the *Thompson sampling* algorithm of [3] (denoted as THOMPSON), and the *Golden Ratio Search* algorithm of [19] (denoted as GRS). Note that both UCB and THOMPSON proposed in [2, 3] were initially designed for the *capacitated* MNL model, in which the number of items each assortment contains is restricted to be at most $K < N$. In our experiments, we operate both the UCB and THOMPSON algorithms under the uncapacitated setting, simply by removing the constraint set when performing each assortment optimization.

Most hyper-parameters (such as constants in confidence bands) are set directly using the theoretical values. One exception is our LIL-TRISECT algorithm, in which we remove the coefficient of 4 in front of the square root term in the confidence bands in Eq. (6), which can be thought of as taking $\varepsilon \to 0^+$ in the finite-sample LIL inequality (see Lemma 14) and was also adopted in [16]. Another exception is the GRS algorithm: in [19] the number of exploration iterations is set to $34 \ln(2N)/\beta^2$ where $\beta = \min_{j \neq j'} |R(\mathcal{L}_{r_j}) - R(\mathcal{L}_{r_{j'}})|$, which is inappropriate for our "gap-free" synthetical setting in which $\beta = 0$. Instead, we use the common choice of $\sqrt{T}$ exploration iterations in typical gap-independent bandit problems for GRS.

**Results.** In Table 1 we report the mean and maximum regret from 20 independent runs of each algorithm on our synthetic data, with different settings of $N$ (number of items) and $T$ (time horizon). We observe that as the number of items ($N$) becomes large, our algorithms (TRISEC and LIL-TRISEC) achieve smaller mean and maximum regret compared to their competitors, and LIL-TRISEC consistently outperforms TRISEC in all settings. Unlike UCB and THOMPSON whose regret depend polynomial on $N$, our TRISEC and LIL-TRISEC algorithms have no dependency on $N$ and hence their

regret does not increase significantly with $N$. While GRS also has weak (logarithmic) dependency on $N$, its pure exploration plus pure exploitation structure makes its performance rather unstable, which is evident from the large gaps between mean and maximum regret of GRS.

## 6 Discussion and conclusion

In this paper we consider the dynamic assortment allocation problem under uncapacitated MNL models and derive near-optimal regret bounds. One important open question is to further remove the $O(\sqrt{\log \log T})$ term in the upper bound in Theorem 2 and eventually achieve upper and lower regret bounds that match each other up to universal numerical constants. We conjecture that such improvement is possible by considering a sharper LIL concentration inequality which, instead of holding uniformly for all $t \in \{1, 2, \cdots\}$, holds only at "doubling checking" points $\{1, 2, 4, 8, \cdots\}$.

Other questions worth investigating is to design "horizon-free" algorithms which automatically adapts to the time horizon $T$ that is not known a priori, and "instance-optimal" regret bounds whose regret depends explicitly on the problem parameters $\{r_i\}_{i=1}^n$, $\{v_i\}_{i=1}^n$ and matching corresponding (instance-dependent) minimax lower bounds in which $\{v_i\}_{i=1}^n$ are known up to permutations. Such instance-optimal regret might potentially depend on "revenue gaps" $\Delta_i = R(S^*) - R(\mathcal{L}_{r_i})$, where $S^*$ is the optimal assortment and $r_i$ is the revenue parameter of the item with the $i$th largest revenue.

### Acknowledgments

Xi Chen would like to thank the support from Alibaba Innovation Research Award and Bloomberg Data Science Research Grant. Part of the work was done when Yuan Zhou was visiting the Shanghai University of Finance and Economics.

## Footnotes

[1]The constraint $r_i \leq 1$ is without loss of generality, because it is only a normalization of revenues.

[2]See the related work section 1.2 for details.

[3] By Lemma 2, we have $y_\tau - F(y_\tau) \geqslant y_\tau - F(x_\tau) \geqslant y_\tau - x_\tau$

[4] Stop whenever the maximum number of iterations $T$ is reached.

[5] $\delta = O(1/T^2)$ rather than $\delta = O(1/T)$ is used because an additional union bound is required for all inner iterations $t$ in each outer iteration $\tau$ for confidence intervals constructed via the Hoeffding's inequality.

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
