[Reviews · NeurIPS 2018]

Reviewer 1



The authors consider a problem they call "dynamic assortment", where a learner selects a subset of items S_t amongst a ground set of items [N], and presents it to a customer. The customer chooses item i in S_t with probability proportional to v_i (an unknown parameter), and the learner receives a reward of r_i. The authors present an algorithm and prove guarantees on its regret (in particular that it is independent of the number of items). The paper is very well written. While the algorithm is mostly a combination of well known techniques in the continuous bandit literature (bisections, trisections etc.), the authors resort to some nice tricks to reduce regret. - In section 3, the authors prove that, in order to solve the problem, it is sufficient to solve a one dimensional problem on [0,1], where one must maximize a piecewise constant function (or in fact find its fixed point). This problem has been addressed in https://arxiv.org/abs/1604.01999, (both in the adversarial and stochastic context), so that the authors could compare their results to this. - The trick used to reduce the regret from \sqrt{T \ln T} to \sqrt{T \ln \ln T} is very nice, and it begs the question whether or not this can be applied to other "generalized" trisection algorithms such as those studied by https://arxiv.org/abs/1107.1744 , https://arxiv.org/abs/1406.7447 and http://www.icml-2011.org/papers/50_icmlpaper.pdf - as a minor remark, in equation (1), \min should be \max (the optimal arm MAXIMIZES the revenue)

Reviewer 2



Summary: In the paper, the authors consider an online assortment problem, where each customer chooses his/her purchase from the offered assortment according to a multinomial logit choice model. The preference weights in the choice model are unknown to the seller, who aims to maximize the total revenue under the model uncertainty. The authors consider the unconstrained setting, which allows the seller to offer any subset of products as the assortment. This is different from most online assortment optimization literature, which consider the constrained setting where there is a cardinality constraint on the number of products in an assortment. The authors first note the fact that, in the unconstrained setting, the optimal assortment lies in the sequence of revenue-ordered assortments. Using the established facts that the sequence of revenue-ordered assortments is nested and the sequence of mean revenue associated with the sequence of assortments, the authors transform the assortment optimization problem to a certain unimodal bandit problem. The authors accomplish a O(sqrt{T}log log T) regret bound, where T is the number of rounds, and the bound is independent of the number of products. Major Comments: While online assortment optimization is an important problem in the realm of e-commerce, I am not too convinced if the submission is sufficient for publication. Indeed, cardinality constraints are an important aspect in the e-commerce setting, since the seller can only display a limited number of products out of the set of all products, and the set is typically much larger than the display capacity. Therefore, with the absence of cardinality constraints, the submission shall be mainly weighted in terms of its theoretical contributions. Given that the transformation to a unimodal optimization is a well-known result in the revenue optimization literature (for example, see the classical papers [TalluriVR04, RusmevichientongT12]), the main theoretical result shall be on addressing the unimodal bandit problem in the case of assortment optimization. My major concern about the work is my difficulty in comparing the submission with existing works on unimodal bandits. While the submission claims (in Lines 96-97) that existing unimodal bandit literature requires structural assumptions such as convexity and inverse Lipschitz continuity, to my knowledge the design and analysis of Algorithm OSUB in [CP14] (see Sections 4.2, 4.3) seem not require any such sort of structural assumption. In fact, [CP14] seems to only require unimodality, same as the current paper. [CP14] provides a “gap-dependent” regret bound, while the current paper provides a “gap-independent” regret bound, where “gap” here refers to the difference between the optimal reward and a suitably defined second best reward. Understandably, we cannot directly compare a gap-independent regret bound with a gap-dependent one. Thus, we cannot conclude that the proposed algorithm in the submission achieves a better bound than Algorithm OSUB. In relation to the previous comments, for assessment it will be beneficial for the authors to provide concrete examples (say in the supplementary material) on showing that structural assumptions such as convexity and inverse Lipchistz continuity (I believe that the literature are essentially [AFHKR13, YM11]) do not hold in the current problem setting. (Post Rebuttal) I am convinced of the paper's technical novelty after the author's feedback. I feel that the improvement from log t to loglogt somehow overcome the models' limitation on the lack of capacity constraint. Minor Comments: It shall be interesting to numerically compare the proposed algorithms with existing unimodal bandit algorithms [CP14, YM11] (in particular Algorithm OSUB in the former), and investigate if the proposed algorithms indeed suit better for the unconstrained assortment optimization problem empirically. References [AFHKR13] A. Agarwal, D. P. Foster, D. Hsu, S. M. Kakade, and A. Rakhlin. Stochastic convex optimization with bandit feedback. SIAM Journal on Optimization, 23(1):213–240, 2013. [CP14] R. Combes and A. Proutiere. Unimodal bandits: Regret lower bounds and optimal algorithms. In ICML, 2014. [RT12] P. Rusmevichientong and H. Topaloglu. Robust assortment optimization in revenue management under the multinomial logit choice model. Operations Research, 60(4):865–882, 2012. [TVR04] Talluri, K., and G. J. van Ryzin. 2004. Revenue management under a general discrete choice model of consumer behavior. Management Science 50 (1): 15–33. [YM11] J. Y. Yu and S. Mannor. Unimodal bandits. In ICML, 2011.

Reviewer 3



This paper provides two algorithms for the dynamic assortment planning problem with the multinomial-logit probabilistic model. It is well written and clearly conveys the ideas. The proofs in the supplementary are also very clear, with nice order. In my opinions, the contribution of this paper mainly falls into two parts: 1. An upper bound independent of item number N. The upper bound of worst-case regret of these two algorithms are both independent of the item number N, while in the previous work on this problem at least logN will appear. This is a very important and also surprising progress in dynamic assortment planning problems. It can greatly improve the algorithm performance in practice in cases when N is large, e.g., online sales and advertisement. The numerical experiment in the paper also shows very good and stable performance, compared with previous algorithms, because of the aforementioned better upper bound independent of N. 2. Removing logT and using loglogT instead. The second algorithm (LIL-trisection) in this paper, using Hoeffding’s inequality, improves the worst-case upper bound of regret to O(sqrt{TloglogT}) from O(sqrt{TlogT}). This is also highly nontrivial. Since the paper in the meanwhile gives a lower bound for the worst-case regret by O(sqrt{T}), the new loglogT bound goes a big step towards the best case. The paper also conjectures that the worst-case regret actually has the same upper and lower bound up to a constant. The small gap left here makes the target of further research quite clear. Some other comments on the paper (Please correct me if any of my comments is wrong): The paper transfers the search for the optimal assortment to the search for the maximizer of the revenue potential function, which is the expected revenue of a “level-set” type assortment. This step is very beautiful, and the paper gives a very detailed description of the revenue potential function and proves several useful properties that can benefit further research. For the experiment part, I am curious why all r_i’s are constrained within [.4, .5] while they are initially assumed to be normalized to [0, 1]. This point might be trivial, but some explanations of why it is more convenient, or consistency of the numbers are recommended.